# Majorana Fermions in One-Dimensional Structures at LaAlO₃/SrTiO₃ Oxide Interfaces

**Maria Vittoria Mazziotti [1], Niccolò Scopigno [2], Marco Grilli [3,4] and Sergio Caprara [3,4,*]**

[1] Dipartimento di Matematica e Fisica, Università Roma Tre, Via della Vasca Navale 84, 00146 Rome, Italy; mariavittoria.mazziotti@unitoma3.it

[2] Institute for Theoretical Physics, 3584 CC Utrecht, The Netherlands; n.scopigno@uu.nl

[3] Dipartimento di Fisica, Università di Roma Sapienza, Piazzale Aldo Moro 5, 00185 Rome, Italy; marco.grilli@roma1.infn.it

[4] CNR-Istituto dei Sistemi Complessi, Via dei Taurini 19, 00185 Rome, Italy

[*] Correspondence: sergio.caprara@roma1.infn.it; Tel.: +39-06-4991-4294

**Abstract:** We study one-dimensional structures that may be formed at the LaAlO₃/SrTiO₃ oxide interface by suitable top gating. These structures are modeled via a single-band model with Rashba spin-orbit coupling, superconductivity and a magnetic field along the one-dimensional chain. We first discuss the conditions for the occurrence of a topological superconducting phase and the related formation of Majorana fermions at the chain endpoints, highlighting a close similarity between this model and the Kitaev model, which also reflects in a similar condition the formation of a topological phase. Solving the model in real space, we also study the spatial extension of the wave function of the Majorana fermions and how this increases with approaching the limit condition for the topological state. Using a scattering matrix formalism, we investigate the stability of the Majorana fermions in the presence of disorder and discuss the evolution of the topological phase with increasing disorder.

**Keywords:** Majorana fermions; topological superconductors; Rashba spin-orbit coupling; oxide interfaces

## 1. Introduction

The quest for Majorana fermions (MFs) is one of the hottest topics in condensed matter physics [1–3]. MFs are completely neutral fermion particles that coincide with their own antiparticles and have been predicted to exist as an emergent quasi-particle zero-energy state in a very special class of materials: the topological superconductors. Besides the fundamental aspects of MFs, their non-Abelian braiding properties offer possible application in quantum computation [4–6].

Roughly speaking, the existence of MFs relies on two main ingredients: the possibility to mix on equal footing electron and hole states to form a neutral object (as happens in superconductors) and the (effective) lack of spin degrees of freedom, because, otherwise, the standard Bogoljubov combination of electron and hole would not coincide with its own antiparticle (see, e.g., [1]).

From the theoretical point of view, the simplest model that captures these two ingredients, and indeed hosts MFs, is the Kitaev chain [4]. It describes spinless electrons hopping between the sites of a one-dimensional (1D) tight-binding chain, hosting superconducting pairing with *p*-wave symmetry. Kitaev observed that, varying the model parameters, the system can be in two distinct phases, one said to be trivial, the other topological. If we decompose the fermion operators on each site into their real and imaginary parts, $c_x = \gamma_{x,1} + \iota\, \gamma_{x,2}$ and identify the $\gamma$ operators with MFs, then in the trivial phase, MFs are bound into ordinary fermions only. In the other phase, almost all MFs bind to form ordinary fermions, while two MFs are left unpaired at both ends of the chain. Of course,

for generic values of the parameters in the topological phase, the MFs are not localized on each single end-site, but extend somewhat inside the chain with an exponential tail.

The key obstacle standing in the way of the physical realization of Kitaev's paradigm is the electron spin. In most natural realizations of a 1D chain, the electron spin is present and must somehow become a quenched degree of freedom [7–9]. In this regard, the spin-orbit coupling is appealing because it is a rather common mechanism removing the spin degeneracy. With the help of a magnetic field, one can then hope to freeze the degrees of freedom higher in energy, achieving an effective spinless fermion system.

From the experimental point of view, there have been a number of proposals to engineer and detect MFs in two-dimensional (2D) solid-state systems, including the fractional quantum Hall liquids [6], interacting quantum spin systems [10], spin-polarized *p*-wave superconductors [11] and, more recently, interfaces between topological insulators [12] or semiconductors [1,5,13] and ordinary superconductors. 1D geometries may provide a more direct environment for establishing and detecting MFs, since these are expected to be localized at the ends of the wire or at the interface between the trivial and topological regions of the system. Moreover, the expected scarceness of low-energy excitations, which could interfere with the detection of MFs, is another advantage of 1D geometries. Ferromagnetic atomic chains [14] or semiconducting nanowires [15,16] proximized with *s*-wave superconductors have been explored.

In this paper, we analyze the possibility to observe the MFs in oxide heterostructures, especially at the $LaAlO_3$/$SrTiO_3$ (LAO/STO) interfaces, since these host at the same time *s*-wave superconductivity [17–27] and strong Rashba spin-orbit coupling (RSOC) [28–32]. Therefore, by adding a magnetic field, one has all the ingredients to achieve the quenching of the spin degrees of freedom and topological superconductivity [33,34] in one and the same system.

In particular, we consider the possibility to manipulate the LAO/STO 2D electron gas to produce a 1D structure. Recent advances in top gating of these systems [32] open the way to achieving this goal in the near future. In this case, one would deal with a system where an effective Kitaev-like model could be realized. The scope of the present paper is precisely to investigate a theoretical 1D model apt to describe the occurrence of MFs in the 1D gas formed at the LAO/STO interface. Moreover, and quite importantly, the gate permits a fine tuning of the electron density, allowing for a control of the chemical potential, to access and exit the topological phase of the superconducting state.

This paper is organized as follows: Section 2 is devoted to the presentation of the model. In Section 3, the model is analyzed in reciprocal space, while its real space analysis in carried out in Section 4. Sections 3 and 4 are rather pedagogical and aim at introducing the notation and concepts that will be exploited in Section 5, where the effect of disorder is studied in real space, providing the main results of this piece of work. Our concluding remarks can be found in Section 6.

## 2. The Model

The multi-band structure of the confined 2D electron gas at the LAO/STO interface is well established, both experimentally [35] and theoretically [36–38]. In particular the 2D confinement along the *z* direction (perpendicular to the interface) has two main consequences. On the one hand, the Ti $d_{xy}$ orbitals, which would form less dispersive bands in the confined direction, give rise to more narrowly-spaced discrete levels from which the continuous bands in the *x*, *y* directions depart. On the other hand, the other Ti orbitals ($d_{xz}$ and $d_{yz}$) would form more dispersive bands along *z* and therefore form more spaced sub-bands starting at a higher energy with respect to the $d_{xy}$ sub-bands. Thus, the confinement along *z* over distances of 5–20 nm significantly alters the electronic structure, as well as the electronic properties of the system. For instance, it has been shown that the 2D electron gas at the LAO/STO interface acquires a large compressibility [30,31,39–42]. Together with the steadily improving capability of controlling the top gating [32], this large compressibility opens the way to the possibility of designing and engineering configurations in which the 2D electron gas is confined in a 1D geometry, as schematized in Figure 1.

However, similarly to the strong changes undergone by the bulk STO under confinement along $z$, it is obvious that a further confinement into a 1D geometry will have a conspicuous impact on the electronic structure and properties of the LAO/STO interface, which requires a careful study that we leave outside the scope of the present work. For the sake of simplicity, we shall assume that shrinking the system along $y$ produces a 1D electron gas along $x$, increasing the energy spacing between the lowest $d_{xy}$ sub-bands (which will also raise in energy). This will leave one low-energy non-degenerate sub-band with a rather light mass along $x$ (we are assuming that the confinement along $y$ is over distances larger than those along $z$, otherwise the $d_{xz}$ and $d_{xy}$ sub-bands would become degenerate). Therefore, we generically expect a one-band description of superconductivity to be appropriate, contrary to the case of the two-dimensional LAO/STO interface, where multi-band superconductivity may occur [25,43]. We shall describe the 1D electron gas within a single-band tight-binding model, as a chain of length $L$ with $N$ sites, separated by a lattice spacing $a$.

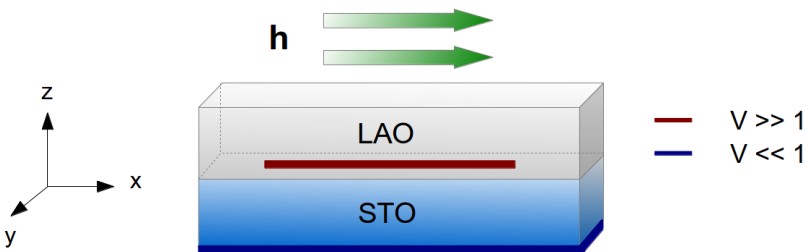

**Figure 1.** Applying a suitable gate voltage, $V$, to the LAO/STO heterostructure, one can create a 1D electron gas at the interface. We can assume that in a back-gating configuration with large negative back gating (labeled by $V \ll 1$), an insulating state is generically present at the interface, while the subsequent introduction of a 1D-like large positive top-gate potential (labeled by $V \gg 1$) creates a locally conductive region [40,44].

In the grand-canonical ensemble, our 1D chain, including an RSOC, a magnetic field and a superconducting order parameter, is described in reciprocal space by the Hamiltonian:

$$
\begin{aligned}
H = \sum_{k,\sigma,\sigma'} & \left( \xi_k \tau^0_{\sigma,\sigma'} - \eta_k \tau^y_{\sigma,\sigma'} - h_x \tau^x_{\sigma,\sigma'} \right) c^\dagger_{k,\sigma} c_{k,\sigma'} \\
& + \sum_k \left( \Delta c^\dagger_{k,\uparrow} c^\dagger_{-k,\downarrow} + h.c. \right),
\end{aligned}
\tag{1}
$$

where $\xi_k \equiv -2t\cos(ak) - \mu$, with $k \in [-\pi/a, \pi/a]$ and $a \approx 3.9\,\text{Å}$, is the band dispersion law, with respect to the chemical potential $\mu$, $t$ is the hopping, and $\eta_k \equiv 2\alpha \sin(ak)$ and $h$ are the RSOC term and the magnetic field, respectively, with $h$ directed along the chain ($x$). Finally, $\Delta$ is the $s$-wave superconducting order parameter, while $\tau^u_{\sigma,\sigma'}$ ($u = 0, x, y, z$) are the Pauli matrices. Our model describes a gate-induced "nanowire" geometry, which also takes into account the peculiarities of the electron gas at the LAO/STO interface, namely Rashba spin-orbit coupling and an intrinsic $s$-wave superconductivity, which therefore does not require any additional proximity-induced superconductivity to support topological states.

At low electron density, the tight-binding bands are filled only near the $\Gamma$ point and can be described by a quadratic dispersion. The relation between mass and hopping reduces to $t = \hbar^2/(2\,ma^2)$. The two effective masses, measured by Santander et al. [45], $m_L \approx 0.7\,m_e$ for the planar direction and $m_H \approx 20\,m_e$ for the out-of-plane direction (where $L$ and $H$ stands for light and heavy, respectively) yield $t_L \approx 0.36\,\text{eV}$ and $t_H \approx 0.0125\,\text{eV}$. Under confinement along $y$, we assume that the resulting 1D electronic structure inherits the light mass along $x$, and we consider a hopping parameter $t \simeq 0.36\,\text{eV}$ along $x$. Henceforth, we work in units of the lattice spacing $a$ and of the hopping parameter $t$.

Recent magnetoconductivity experiments on LAO/STO interfaces [28,46] find substantial values for the RSOC, $\approx [10^{-12} - 10^{-11}]$ eV·m. From this value, we can deduce $\alpha = [10^{-2} - 10^{-1}]t$. As far as the typical intensity of the Zeeman field is concerned, several works [27,47] show that, using in-plane magnetic fields, superconductivity persists up to $h \approx 2.2\,\text{T} \simeq 10^{-3}\,t$. To fix the order of magnitude of the superconducting order parameter, we use the Bardeen-Cooper-Schrieffer relation between $\Delta$ and the critical temperature $T_c$,

$$\frac{2\Delta}{k_B T_c} \simeq 3.4 \implies \Delta = 1.7\, k_B T_c \simeq 10^{-4}\,t,$$

where a superconducting critical temperature $T_c \sim 200\,\text{mK}$ has been assumed. We also assume that superconductivity along $x$ is not substantially affected by confining the electron gas along $y$ over distances smaller than (but of the order of) the superconducting coherence length $\xi \sim 50\,\text{nm}$. Of course, we are aware that a strictly 1D electron gas can no longer sustain superconductivity, but we assume that the physics we are interested in occurs over length scales (the length of our 1D chain, $L$) smaller than the superconducting coherence length along the chain.

Figure 2 shows the effect of the different contributions to the Hamiltonian (1) on the spectrum. The RSOC, resulting in a sort of effective momentum-dependent magnetic field, separates electrons with opposite spins in momentum space. The blue (red) parabola in Figure 2a, corresponds to electronic states whose spin aligns along the positive (negative) $y$ axis. Clearly, no spinless regime is possible here, and the spectrum always supports an even number of pairs of Fermi points for any $\mu$. The introduction of a uniform magnetic field, parallel to the direction of motion of the carriers, explicitly breaks the time-reversal symmetry by removing the spin degeneracy at $k = 0$, and two pseudo-spin bands are formed with a Zeeman-induced gap $\sim 2h$. Since one band (red in Figure 2) may be pushed above the chemical potential, an effective spinless regime results, which is necessary to realize a topologically non-trivial state. In order to completely realize an effective Kitaev-like model, we just need to introduce a superconducting order parameter whose effect is to modify the Zeeman gap at $k = 0$ and to open a gap at the Fermi momenta $\pm k_F$; see Figure 2c.

In order to clarify how the presence of a superconductivity is reflected in an unconventional Cooper pairing within such a spinless regime, we can rewrite the Hamiltonian (1) in terms of the operators $c_{k,\pm}^\dagger$ that creates electrons with energy $\varepsilon_{k,\pm}$ (dispersion in the absence of superconductivity). In this basis, the Hamiltonian reads:

$$\begin{aligned}
H = &\sum_{k,\lambda=\pm} \varepsilon_{k,\lambda} c_{k,\lambda}^\dagger c_{k,\lambda} \\
&+ \sum_k \left( \Delta_{+,-} c_{k,+}^\dagger c_{-k,-}^\dagger + h.c. \right) \\
&+ \sum_k \left( \Delta_{+,+} c_{k,+}^\dagger c_{-k,+}^\dagger + \Delta_{-,-} c_{k,-}^\dagger c_{-k,-}^\dagger + h.c. \right),
\end{aligned} \qquad (2)$$

with $\varepsilon_{k,\pm} \equiv \xi_k \pm \zeta_k$, $\zeta_k \equiv \sqrt{h^2 + \eta_k^2}$,

$$\Delta_{+,-} \equiv \frac{h}{\zeta_k} \Delta,$$

$$\Delta_{\pm,\pm} \equiv \mp \frac{\iota \eta_k}{2\zeta_k} \Delta.$$

The first line of Equation (2) describes the band energies, while the second describes interband *s*-wave pairing. Most importantly, the third line encodes intraband *p*-wave pairing. This emerges because, as shown schematically in Figure 2c, the electrons at $\pm k_F$ in a given band have misaligned spins, and they can form Cooper pairs in response to $\Delta$. Due to Fermi statistics, the effective potential $\Delta_{\pm,\pm}$ that pairs these electrons must exhibit odd parity that reflects the spin-flip of electron spins as one sweeps the momentum from $k$ to $-k$. As will be shown below, and similarly to the original Kitaev model, this ensures that the 1D electron gas may be found in a topological phase that supports MFs

at the interface with a trivial-state like, e.g., the vacuum or the 2D insulating state surrounding the metallic 1D chain at the LAO/STO interface.

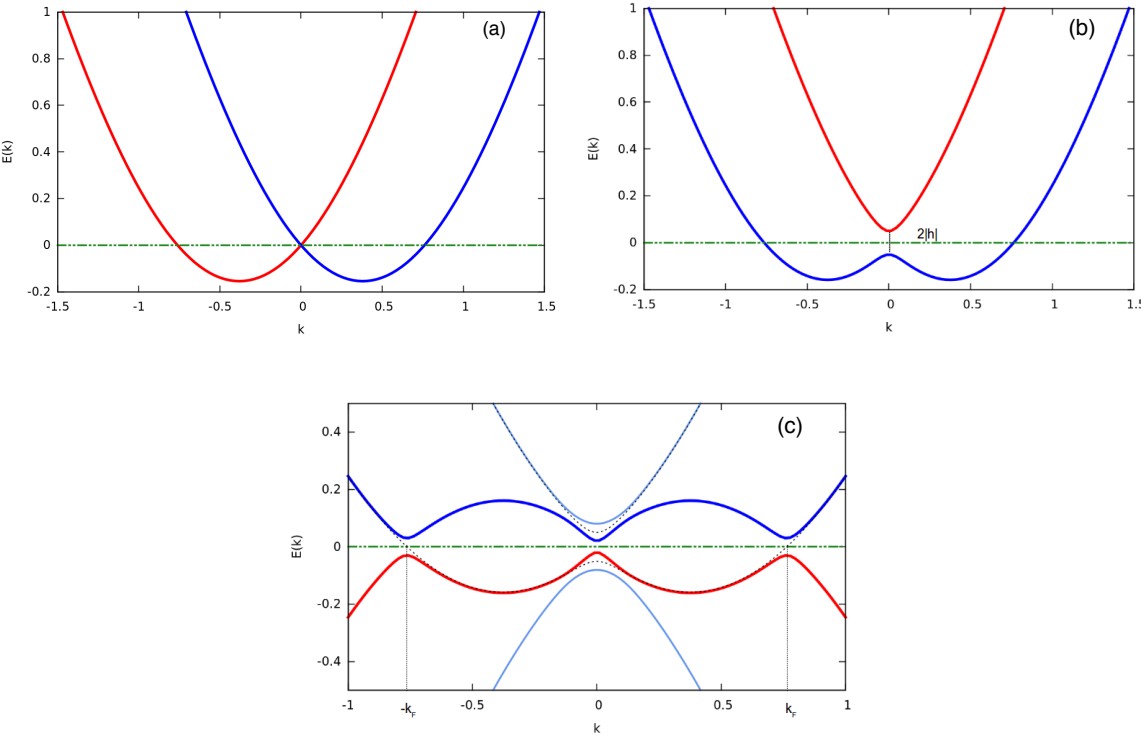

**Figure 2.** The Hamiltonian spectrum, Equation (1), for $a = 1$, $\alpha = 0.4\,t$, $\mu = -2t$, $h = 0.09\,t$ and $\Delta = 0.04\,t$ ($t = 1$). Panel (**a**) shows electronic bands split by Rashba spin-orbit coupling (RSOC). Opposite spin directions along the Rashba effective field ($y$ axis) are denoted in blue (spin aligned along $-y$) and red (spin along $+y$). The dotted green curve is, instead, the rescaled chemical potential $\tilde{\mu} = \mu + 2t$. Panel (**b**) shows the Hamiltonian spectrum when a Zeeman field is applied parallel to the $x$ direction, giving rise to a gap $2|h|$ at $k = 0$. Finally, Panel (**c**) shows the effect of a superconducting gap. Superconductivity opens a gap at the Fermi points, $\pm k_F$, and reduces the Zeeman gap at $k = 0$.

We notice here that our model shares some similarities with a previous model introduced in [48] for a single-band two-dimensional electron gas. Although the authors of [48] assumed the presence of ferromagnetic moments as a property of the LAO/STO interface, their effect is analogous to the externally-applied magnetic field along $x$, which we consider in the present work. However, due to our one-dimensional geometry, we do not need an extra magnetic field along $z$ to push one of the chiral states to higher energy, thereby reaching an effective spinless state. Finally, despite some similarity in the expressions for $\Delta_{+,-}$ and $\Delta_{\pm,\pm}$, the different roles of the magnetic fields entail different conditions for the occurrence of topological superconductivity, as we discuss in Section 3.

## 3. Description of the Topological Phase in Reciprocal Space

In order to determine the conditions under which the model introduced in the previous section can support MFs at zero energy, it is instructive to first explore the bulk properties of the chain, which can be conveniently studied by imposing periodic boundary conditions on the system.

The Hamiltonian (1) can be written in the usual Nambu basis $\Psi_k^\dagger = [c_{k\uparrow}^\dagger, c_{k\downarrow}^\dagger, c_{-k\downarrow}, c_{-k\uparrow}]$ in the standard Bogoljubov–de Gennes form:

$$H = \frac{1}{2} \sum_k \Psi_k^\dagger H_k \Psi_k, \tag{3}$$

where:

$$H_k = \begin{pmatrix} \xi_k & \iota\eta_k - h & \Delta & 0 \\ -\iota\eta_k - h & \xi_k & 0 & -\Delta \\ \Delta & 0 & -\xi_k & \iota\eta_k + h \\ 0 & -\Delta & -\iota\eta_k + h & -\xi_k \end{pmatrix}. \tag{4}$$

Then, from Equation (4), we obtain the bulk spectrum:

$$E_{k,\lambda,\nu} = \lambda\sqrt{\Delta^2 + \xi_k^2 + \zeta_k^2 + \nu\varphi_k^2}, \quad \lambda, \nu = \pm 1, \tag{5}$$

where $\varphi_k^2 = 2\sqrt{\zeta_k^2\xi_k^2 + \Delta^2 h^2}$.

The transition to the topological state occurs only when the gap of the bulk spectrum closes, i.e., when

$$h = \sqrt{\Delta^2 + (\mu + 2t)^2}, \tag{6}$$

which reduces to the Kitaev relation $\sqrt{\Delta^2 + \tilde{\mu}^2}$ with $\tilde{\mu} \equiv \mu + 2t$.

As we show in Figure 3a, the gap in the low-energy spectrum closes for values of $h$ that obey Equation (6). This reflects the *p*-wave nature of the pairing required by Pauli exclusion: since $\Delta_{\pm,\pm}$ is an odd function of $k$, Cooper pairing at $k = 0$ or $k = \pm\pi$ is prohibited. The situation changes for the high-energy spectrum, which stays always gapped, as shown in Figure 3b. Therefore, the condition for the system to host MFs is only determined by the spectrum of the low-energy excitations.

With respect to the low energy spectrum, for $h = \sqrt{\Delta^2 + \tilde{\mu}^2}$, the system is found in a gapless state that separates states with different superconducting pairing. The two phases have the same physical symmetries, and the transition between them has a topological character.

*Topological Criterion*

There are several ways in which one can express the topological invariant distinguishing the two gapped phases that we mentioned above, like, e.g., the Chern number, allowing one to distinguish the various phases in 2D quantum Hall systems and the more recently proposed $Z_2$ invariants that characterize topological insulators in two and three dimensions. For 1D topological superconductors, the relevant topological invariant is the Majorana number, $M = \pm 1$, first introduced by Kitaev. In his seminal paper [4], Kitaev showed that all 1D fermion systems with superconducting order fall into two categories distinguished by $M$. The presence of unpaired MFs is hallmarked by $M = -1$, and the system is gapped.

In order to calculate the Majorana number, we can follow different approaches. Here, we shall analyze the approach proposed by Kitaev, while an alternative method is presented in Appendix A.

A Hamiltonian for any non-interacting translationally-invariant fermion system in 1D can be written in the Majorana representation as:

$$H = \frac{\iota}{4} \sum_{l,m,\alpha,\beta} B_{\alpha,\beta}(l-m)\gamma_{l,\alpha}\gamma_{m,\beta}, \tag{7}$$

where $l$, $m$ denote the lattice sites, while $\alpha$, $\beta$ label all other quantum numbers, including spin and orbital degrees of freedom, and the $\gamma$'s are Majorana operators. The Majorana number is defined as:

$$M = \text{sgn}\left\{\text{Pf}\left[\tilde{B}(0)\right]\text{Pf}\left[\tilde{B}(\pi)\right]\right\},$$

where $\tilde{B}(k)$ denotes the spatial Fourier transform of $B(l-m)$ viewed as a matrix with respect to the indices $\alpha$, $\beta$, and Pf$[A]$ denotes the Pfaffian (the square root of determinant with a definite sign) of an antisymmetric matrix. For $k = 0, \pi$, the matrix $\tilde{B}(k)$ is indeed antisymmetric (this follows from the

requirement that the Hamiltonian (7) is Hermitian). The topological invariant for a 1D superconductor can be therefore easily computed directly from the Hamiltonian.

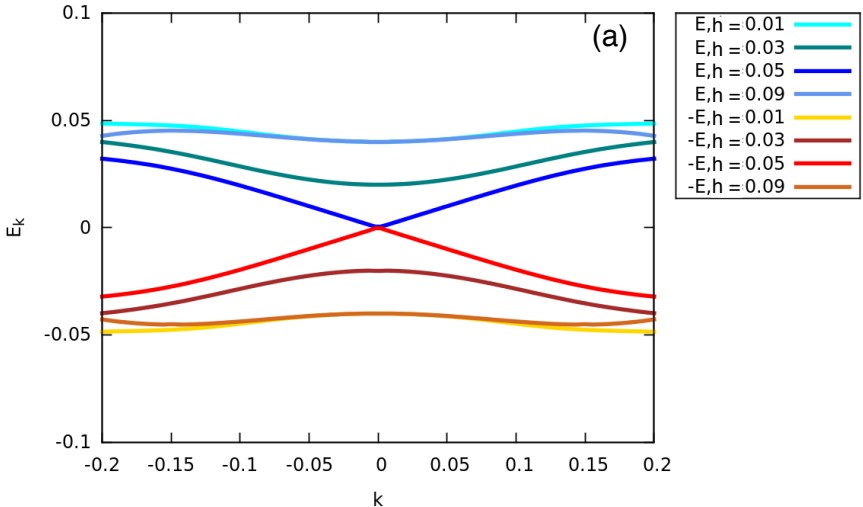

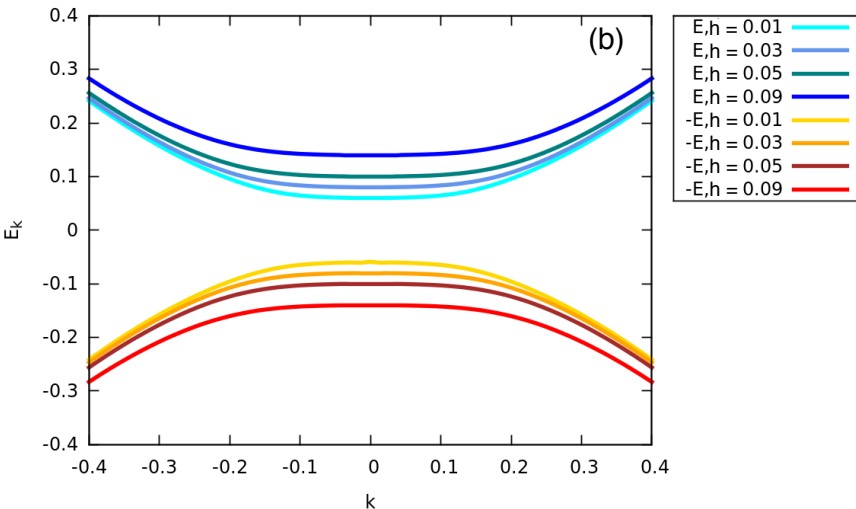

**Figure 3.** (**a**) Pairs $(-E, +E)$ of low-energy levels in the spectrum of Equation (3) as a function of $k$ for $t = 1$, $\mu = -2t$, $\alpha = 0.1\,t$ and $\Delta = 0.05\,t$ at various values of the magnetic field $h$. Here, we have indicated with $E_k$ the lowest energy eigenvalues, i.e., the $E_{k,\lambda,-}$ of Equation (5). (**b**) Same as (**a**), but for the pairs of higher-energy levels of the Hamiltonian. $E_k$ denotes the eigenvalues $E_{k,\lambda,+}$ of Equation (5).

In order to reduce the Hamiltonian (4) to $\tilde{B}(k)$, we decomposed the fermion operators defining the Nambu spinor in Equation (3) in their real and imaginary parts,

$$c_k = \frac{1}{2}(\gamma_{k1} - \iota\,\gamma_{k2}), \quad c_k^+ = \frac{1}{2}(\gamma_{k1} + \iota\,\gamma_{k2}),$$

where the $\gamma$'s are fermion operators that satisfy the relation $\gamma_k^\dagger = \gamma_{-k}$. This yields:

$$H = \iota \sum_k \tilde{\Gamma}_k \tilde{H}_k \tilde{\Gamma}_k,$$

where $\tilde{\Gamma}_k = (\gamma_{k1,\uparrow}, \gamma_{k2,\uparrow}, \gamma_{k1,\downarrow}^\dagger, \gamma_{k2,\downarrow}^\dagger)$, while

$$\tilde{H}_k = \begin{pmatrix} 0 & -\xi_k & 0 & B_k \\ \xi_k & 0 & D_k & 0 \\ 0 & -D_k^* & 0 & -\xi_k \\ -B_k^* & 0 & \xi_k & 0 \end{pmatrix},$$

with $B_k = -\iota \eta_k + h + \Delta$ and $D_k = \iota \eta_k - h + \Delta$. For $k = 0, \pi$ $\eta_k = 0$, and $\tilde{H}_k$ is an antisymmetric matrix, whose Pfaffian is $\text{Pf}[\tilde{H}_{k=0,\pi}] = \xi_{k=0,\pi}^2 - h^2 + \Delta^2$. Therefore, we reduced the problem of determining the topological invariant to the study of $\text{sgn}(\tilde{\xi}_0 \tilde{\xi}_\pi)$, where $\tilde{\xi}_0 = \sqrt{\xi_{k=0}^2 + \Delta^2} - |h| = \sqrt{(2t + \mu)^2 + \Delta^2} - |h|$ and $\tilde{\xi}_\pi = \sqrt{\xi_{k=\pi}^2 + \Delta^2} - |h| = \sqrt{(2t - \mu)^2 + \Delta^2} - |h|$. Assuming that $-2t \leqslant \mu < 0$ (the case $0 < \mu \leqslant 2t$ being completely analogous), we have:

- $\text{sgn}(\tilde{\xi}_0 \tilde{\xi}_\pi) > 0$, for $|h| < \sqrt{(2t + \mu)^2 + \Delta^2}$ or $|h| > \sqrt{(2t - \mu)^2 + \Delta^2}$,
- $\text{sgn}(\tilde{\xi}_0 \tilde{\xi}_\pi) < 0$, for $\sqrt{(2t + \mu)^2 + \Delta^2} < |h| < \sqrt{(2t - \mu)^2 + \Delta^2}$.

Accordingly, the system is in the topologically non-trivial phase when:

$$|h| > \sqrt{\tilde{\mu}^2 + \Delta^2} \quad \text{(topological criterion)}, \tag{8}$$

where we recall that $\tilde{\mu} = 2t + \mu$. The non-trivial topology supports Majorana zero-energy modes at the ends of the 1D chain. These are found studying the system in real space, with open boundary conditions.

## 4. Description of the Topological Phase in Real Space

The real space tight-binding Hamiltonian that describes a linear chain of length $L$, with $N = L - 1$ lattice sites at a distance $a = 1$, with open boundary conditions, in which an RSOC pairing is present, together with a magnetic field parallel to the direction of motion, say $x$, and a conventional superconducting order, reads:

$$\begin{aligned} H = &-\mu \sum_{\sigma, x=1}^{N} c_{x,\sigma}^\dagger c_{x,\sigma} - t \sum_{\sigma, x=1}^{N-1} (c_{x,\sigma}^\dagger c_{x+1,\sigma} + h.c.) \\ &+ \alpha \sum_{\sigma,\sigma', x=1}^{N-1} [\iota \tau_{\sigma,\sigma'}^y (c_{x,\sigma}^\dagger c_{x+1,\sigma'} - c_{x+1,\sigma}^\dagger c_{x,\sigma'}) + h.c.] \\ &- h \sum_{\sigma,\sigma', x=1}^{N} \tau_{\sigma,\sigma'}^x (c_{x,\sigma}^\dagger c_{x,\sigma'} + h.c.) \\ &+ \Delta \sum_{x=1}^{N} (c_{x,\uparrow}^\dagger c_{x,\downarrow}^\dagger + h.c.), \end{aligned} \tag{9}$$

which we can compactly rewrite as:

$$H = \Psi^\dagger H_{4N \times 4N} \Psi + \text{const.}$$

where:

$$\Psi^\dagger = [..., c_{x,\uparrow}^\dagger, c_{x,\downarrow}^\dagger, c_{x,\uparrow}, c_{x,\downarrow}, ...]$$

is the $4N$-component Nambu spinor and:

$$H_{4N \times 4N} = \begin{pmatrix} H_1 & H_2 & 0 & 0 & 0 & 0 \\ H_2^T & H_1 & H_2 & 0 & \ddots & \vdots \\ 0 & \ddots & \ddots & \ddots & \ddots & 0 \\ 0 & 0 & \ddots & \ddots & \ddots & 0 \\ \vdots & \vdots & \ddots & H_2^T & H_1 & H_2 \\ 0 & 0 & .. & 0 & H_2^T & H_1 \end{pmatrix} \tag{10}$$

is a $4N \times 4N$ real symmetric matrix, where:

$$H_1 = \begin{pmatrix} -\mu & -h & 0 & \Delta \\ -h & -\mu & -\Delta & 0 \\ 0 & -\Delta & \mu & h \\ \Delta & 0 & h & \mu \end{pmatrix},$$

$$H_2 = \begin{pmatrix} -t & -\alpha & 0 & 0 \\ \alpha & -t & 0 & 0 \\ 0 & 0 & t & \alpha \\ 0 & 0 & -\alpha & t \end{pmatrix}.$$

The numerical diagonalization of the Hamiltonian provides the energy spectrum shown in Figure 4 for the different magnetic fields and for a chain with $N = 400$ lattice sites.

When $h$ fulfills the topological criterion, Equation (8), the system admits a doubly-degenerate zero-energy mode. According to the Kitaev model, the zero-energy modes are MFs located at the edges of the chain. In order to verify this, we can introduce the Majorana basis:

$$c_{x,\sigma} = \frac{\gamma_{2x-1,\sigma} - \imath \, \gamma_{2x,\sigma}}{\sqrt{2}}, \; c_{x,\sigma}^\dagger = \frac{\gamma_{2x-1,\sigma} + \imath \, \gamma_{2x,\sigma}}{\sqrt{2}}.$$

Then, we can rewrite the Hamiltonian (10) in this basis using the unitary transformation $\tilde{H}_{4N \times 4N} = U_{4N \times 4N} H_{4N \times 4N} U_{4N \times 4N}^T$, where the matrix $U_{4N \times 4N}$ is block-diagonal and splits into $N$ equal $4 \times 4$ matrices $U$ such that:

$$\begin{pmatrix} \gamma_{2x-1,\uparrow} \\ \gamma_{2x-1,\downarrow} \\ \imath \, \gamma_{2x,\uparrow} \\ \imath \, \gamma_{2x,\downarrow} \end{pmatrix} = U \begin{pmatrix} c_{x,\uparrow} \\ c_{x,\downarrow} \\ c_{x,\uparrow}^\dagger \\ c_{x,\downarrow}^\dagger \end{pmatrix}, \quad \textit{i.e., } U = \frac{1}{\sqrt{2}} \begin{pmatrix} 1 & 0 & 1 & 0 \\ 0 & 1 & 0 & 1 \\ 1 & 0 & -1 & 0 \\ 0 & 1 & 0 & -1 \end{pmatrix}.$$

We denote by $\gamma_+$ and $\gamma_-$ the two eigenstates related to the zero eigenvalues. One can first notice that in the Nambu formalism, the spectra are symmetric around the chemical potential $\tilde{\mu} = 0$. Therefore, by ordering the eigenvalues in ascending order (and labeling them with an index $i$), the zero-energy eigenvalues only occur for $i = 2N + 1$ and $i = 2N$ (i.e., $i = 800, 801$ in Figure 4). However, in a finite system, the MFs are coupled by an exponentially small overlap between their wave functions [4], as we will explain below. Therefore, the numerically determined eigenstates are symmetric and antisymmetric superpositions of $\gamma_-$ and $\gamma_+$ with a small finite energy difference. If the numerical eigenstates are denoted with $\gamma_S$ and $\gamma_A$, we generate the states that would correspond to MFs as $\gamma_+ = (\gamma_S + \gamma_A)/\sqrt{2}$ and $\gamma_- = (\gamma_S - \gamma_A)/\sqrt{2}$. Finally, we can define the quantities:

$$P_R(i, \pm) = (\gamma_{2i-1,\uparrow,\pm})^2 + (\gamma_{2i-1,\downarrow,\pm})^2,$$

$$P_I(i, \pm) = (\gamma_{2i,\uparrow,\pm})^2 + (\gamma_{2i,\downarrow,\pm})^2,$$

and obtain the square modulus of the wave function associated with the two MFs:

$$|\psi_\pm(x)|^2 = P_R(x, \pm) + P_I(x, \pm) \tag{11}$$

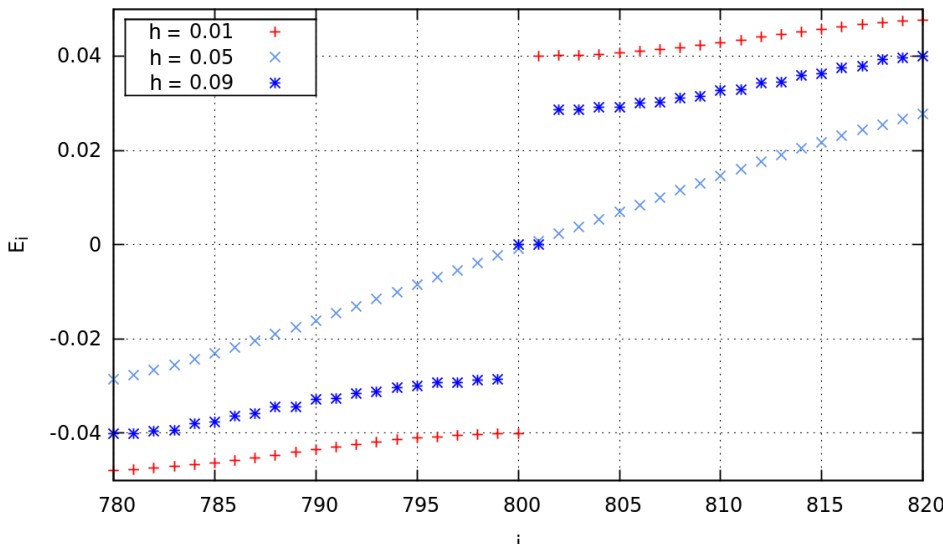

**Figure 4.** The energy spectrum for $h \lesseqgtr \Delta$, $t = 1$, $\mu = -2t$, $\alpha = 0.1\,t$, $\Delta = 0.05\,t$ and $N = 400$; here, $i = 1, ..., 4N$ labels the energy levels in increasing order. For $h > \Delta$, the dispersion relation admits a doubly-degenerate zero-energy mode.

These states are plotted, for $h > \Delta$, in Figure 5, where the red curve is $|\psi_+(x)|^2 \approx P_R(x, +)$, and it is localized at $x = 0$, while the blue curve $|\psi_-(x)|^2 \approx P_I(x, -)$, and it is localized at $x = N$, as expected.

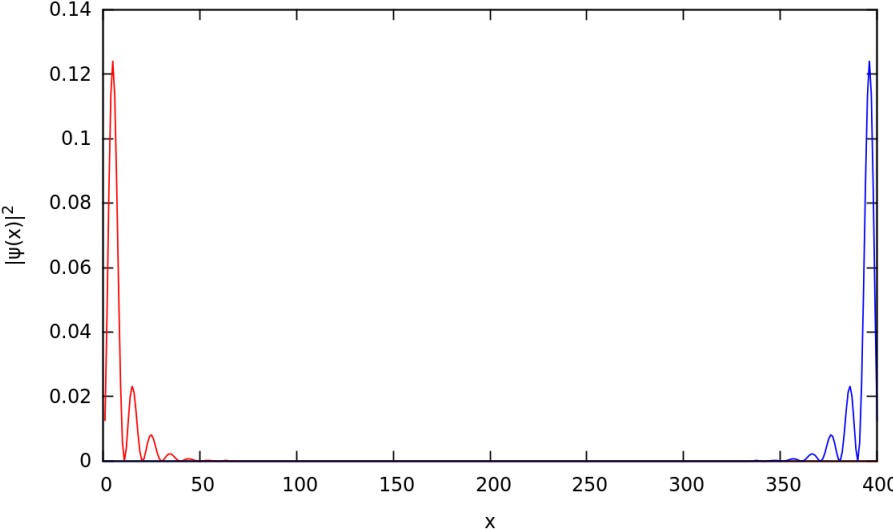

**Figure 5.** The localized Majorana zero-energy modes for, $h = 0.09\,t$, $t = 1$, $\mu = -2t$, $\alpha = 0.1\,t$, $\Delta = 0.05\,t$ and $N = 400$. The figure shows the square amplitude of the wave functions $|\psi_+(x)|^2$ (red curve) and $|\psi_-(x)|^2$ (blue curve). The weak overlap between the wave functions associated with the two Majorana fermions (MFs), not visible in linear scale, depends on the finite length of the chain.

### 4.1. MFs at the Different Chain Lengths

Here, we study the behavior of the wave functions associated with MFs, varying the chain lengths at a fixed value of the magnetic field, $h/t = 0.09 > \Delta/t$. Figure 6 shows the behavior of the wave function for different lengths $L$ of the chain and the corresponding energy spectra. When $L < 100$, the MFs begin to overlap significantly, and the zero-energy level degeneracy is sizeably removed.

Indeed, for a finite chain length, there is a weak mixing of the Majorana zero-energy modes, reflected in a zero-energy level splitting proportional to $e^{-L/\xi}$, where $L$ is the chain length and $\xi$ is a characteristic decay length.

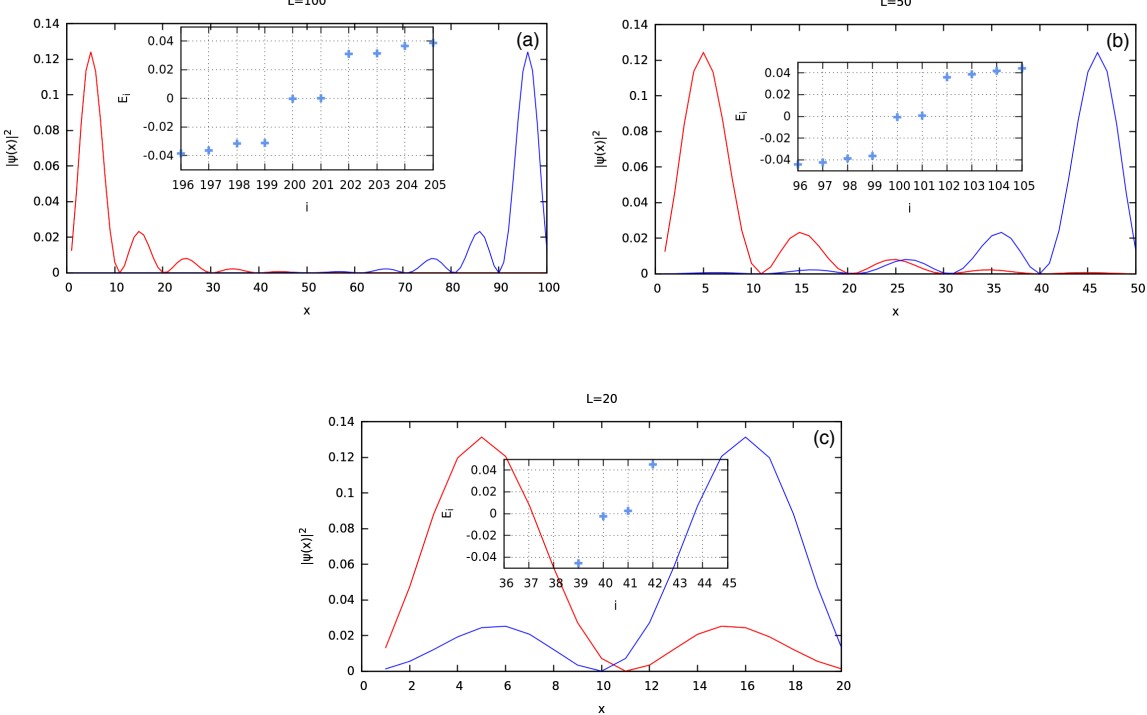

**Figure 6.** The localized Majorana zero-energy modes with varying chain length, for $h = 0.09\,t$, $t = 1$, $\mu = -2t$, $\alpha = 0.1\,t$ and $\Delta = 0.05\,t$. The inset shows the corresponding spectrum. Panel (**a**): $L = 100$; Panel (**b**): $L = 50$; Panel (**c**): $L = 20$.

### 4.2. MFs at Different Magnetic Fields

We can represent the square amplitude of the wave functions associated with MFs as an oscillating function $A(x)$ enveloped by an exponential,

$$|\psi(x)|^2 = A(x)\,e^{-x/\xi}.$$

As shown in Figure 7, the characteristic length scale $\xi$ of the exponential envelope tends to diverge when the field approaches the critical value $h_c = \Delta$.

In this analysis, the quantity of interest is the decay length, $\xi$, vs. $h$. In order to estimate this parameter, we plotted $|\psi_+(x)|^2$ in the semi-logarithmic scale, and we used a linear fitting form for the envelope. We find that for $h \to h_c$, $\xi(h)$ diverges as:

$$\xi(h) \sim \frac{1}{(h - h_c)^\nu}, \quad \text{with } \nu = 1.$$

According to the Kitaev model, when $h$ is varied, the wave functions associated with MFs change gradually from a configuration in which the two MFs, composing an ordinary fermion, couple at adjacent lattice sites (topological phase) to one in which they pair up on the same lattice site (trivial phase). This pictorial view is actually only valid for very specific values of the parameters of the Kitaev model. More generally, the regular fermions are formed mixing in more complicated ways the fermions of the Majorana basis, and in the topological phase, this leads to the formation of MFs with a

finite extension of their wave function. Our analysis shows that this extension gives rise to power-law behaviors similar to those occurring at the ordinary phase transitions [49].

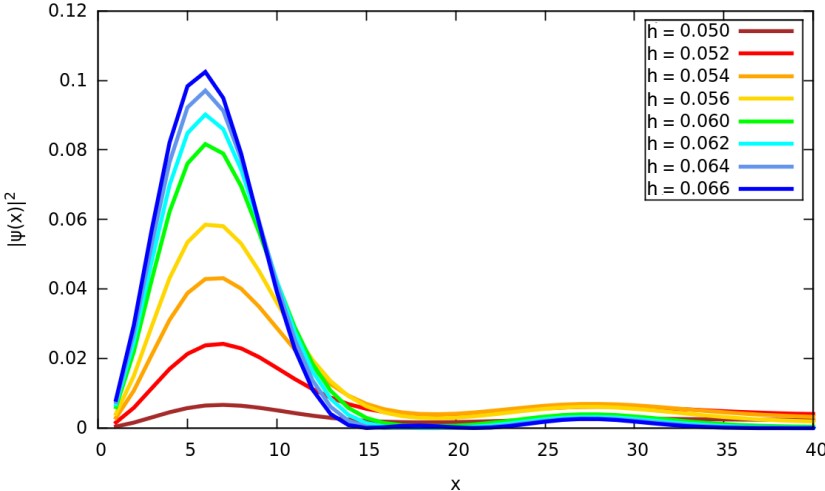

**Figure 7.** The square amplitude of the wave function $|\psi_+(x)|^2$ in the range of $h/t \in [0.05 : 0.06]$, in steps of $\delta h = 0.002\, t$, for fixed $N = 400$.

## 5. The Disordered 1D Chain

In a real system, the presence of defects or impurities makes the crystal lattice not perfectly periodic. In this case, it is preferable to study the system in the real space. However, the classification of topological phases is commonly given in terms of the Hamiltonian of a system with periodic boundary conditions. Therefore, it is necessary to introduce a more general method valid for open boundary conditions. The scattering matrix method, which we summarize in the Appendix A, provides a natural starting point allowing us to study the topological properties of a 1D disordered chain. Recent studies reveal that the MFs are not completely robust against disorder [50–57], and the topological properties are determined by a trade-off between the localizing effects of disorder and the superconducting pairing, which tends to spatially separate the MFs composing a Dirac state.

In this section, we first introduce the 1D disordered chain within a tight-binding model. Then, by means of the scattering matrix method, we determine the topological properties of the system, and we numerically study the robustness of the MFs randomly varying both the local energy level, the hopping and the RSOC. We analyze both the case when the disorder acts on the spin-independent and the case when it acts on spin-dependent terms.

### 5.1. Disordered Tight-Binding Model

The system proposed in the first section can be generalized by assuming that in the Hamiltonian (9), the chemical potential (more precisely, the on-site energy level), and/or the magnetic field, and/or the hopping, and/or the RSOC vary from site to site.

$$
\begin{aligned}
H = & -\sum_{\sigma,\,x=1}^{N} \mu_x c_{x,\sigma}^\dagger c_{x,\sigma} - \sum_{\sigma,\,x=1}^{N-1} t_x (c_{x,\sigma}^\dagger c_{x+1,\sigma} + h.c.) \\
& + \sum_{\sigma,\sigma',\,x=1}^{N-1} [\iota\, \alpha_x \tau_{\sigma,\sigma'}^y (c_{x,\sigma}^\dagger c_{x+1,\sigma'} - c_{x+1,\sigma}^\dagger c_{x,\sigma'}) + h.c.] \\
& - \sum_{\sigma,\sigma',\,x=1}^{N} h_x \tau_{\sigma,\sigma'}^x (c_{x,\sigma}^\dagger c_{x,\sigma'} + h.c.) \\
& + \Delta \sum_{x=1}^{N} (c_{x,\uparrow}^\dagger c_{x,\downarrow}^\dagger + h.c.)
\end{aligned}
\tag{12}
$$

After introducing the Nambu spinor $\Psi_x^\dagger = (c_{x,\uparrow}^\dagger, c_{x,\downarrow}^\dagger, c_{x,\uparrow}, c_{x,\downarrow})$ [50,57], this equation can be rewritten more compactly as:

$$H = \frac{1}{2} \sum_{x=1}^{N} (\Psi_x^\dagger \hat{w}_x \Psi_x + \Psi_x^\dagger \hat{t}_x \Psi_{x+1}). \tag{13}$$

In the matrix representation, the Hamiltonian corresponds to a $4N \times 4N$ matrix, where $\hat{w}_x = \mu_x \tau^0 \tau_N^z - h_x \tau^x \tau_N^z - \Delta \tau^y \tau_N^y$, $\hat{t}_x = -t_x \tau^0 \tau_N^z - \imath \alpha_x \tau^y \tau_N^z$. The Pauli matrices $\tau^{0,x,y,z}$ and $\tau_N^{x,y,z}$ act on the spin and particle-hole spaces, respectively. The Hamiltonian (12) has neither time-reversal, nor spin-rotation symmetry, but it does satisfy the particle-hole symmetry. According to the standard classification of topological states [58], this places the system in the symmetry class $D$, which in one dimension has a topologically non-trivial phase [57,59–61].

To identify the topologically non-trivial phase of the finite disordered chain, it is more efficient to work with the scattering matrix rather than with the Hamiltonian [50,62]. We consider a two-terminal transport geometry, consisting of a disordered superconducting chain of $N$ sites, connected by clean normal-metal leads to reservoirs in thermal equilibrium at zero temperature and chemical potential $\mu = E_F$. Since the system is assumed to be strictly 1D, the leads support four right-moving modes and four left-moving modes at the Fermi level with mode amplitudes $\psi_{in}$ and $\psi_{out}$, respectively (see Figure 8). The presence of four left- and right-moving modes is due to the spin and particle-hole degrees of freedom.

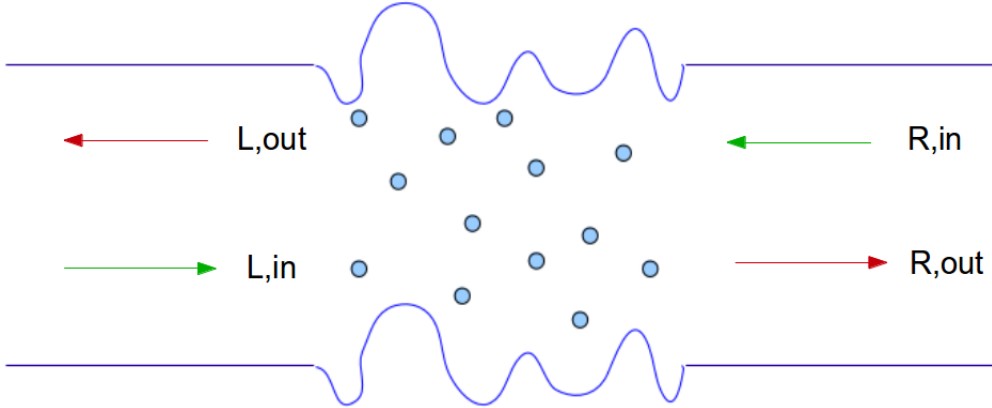

**Figure 8.** Central disordered region connected by means of ideal leads. The modes in the left and right leads are labeled with $L, (in/out)$ and $R, (in/out)$, where the label $in/out$ depends on their propagation direction.

The $8 \times 8$ unitary scattering matrix, $S$, relates incoming and outgoing mode amplitudes:

$$\begin{pmatrix} \psi_{L,out} \\ \psi_{R,out} \end{pmatrix} = S \begin{pmatrix} \psi_{L,in} \\ \psi_{R,in} \end{pmatrix}, \quad S = \begin{pmatrix} r & t' \\ t & r' \end{pmatrix}, \tag{14}$$

where the labels $L$ and $R$ distinguish the modes in the left and right lead. The four blocks of $S$ define the $4 \times 4$ reflection matrices $r$, $r'$ and transmission matrices $t$, $t'$ at the two ends of the chain, respectively. While we give a more detailed description of this technique in Appendix A, here we only recall that the $Z_2$ topological quantum number $Q$, for a system of topological class $D$, is given by [50,57,59–61]:

$$Q = \text{sgn} \, \text{Det} \, (r) = \text{sgn} \, \text{Det} \, (r'). \tag{15}$$

It can be shown [9,63] that the value of $Q$ is determined by the parity of the number Majorana bound states at each end of the chain, with $Q = -1$ in the topological phase. Therefore, the MFs arise at the ends of the chain only when $Q = -1$, and our analysis of the stability of MFs under disorder will precisely involve the calculation of the determinant of the reflection matrix in Equation (15).

*5.2. Results*

Now, we can study the topological properties of the 1D chain described by the Hamiltonian (12). After a direct check of the results obtained in Section 3 for the ordered system (see Figure A1, in Appendix A), we investigate the robustness of MFs when disorder is induced into the on-site energy (Anderson model). Analogous results can be obtained with the same effort, if disorder is introduced into other terms of the Hamiltonian.

When dealing with macroscopic disordered systems, one usually assumes that self-averaging is at work and physical quantities are obtained after the average over the disorder configuration is carried out. However, our aim is to investigate finite length chains of meso- or nanoscopic size. Therefore, owing to the relatively small size of the systems at issue, we find it more informative to work at a fixed disorder configuration. Only when a comparison will be done with quantities often investigated in localization theory (like, e.g., the inverse participation ratio (IPR)), an average over several ($n_{dis}$) disorder configurations will be performed.

To deal with a 1D chain with randomly-assigned on-site energies, we formally assign this randomness to a locally-varying chemical potential, extracted from a uniform distribution in the interval $[\mu - \delta_\mu/2, \mu + \delta_\mu/2]$, where $\mu$ is now the mean value, while $\delta_\mu$ is the fluctuation magnitude.

Figure 9 shows the phase diagram, $\text{Det}(r)$ (Equation (15)), as a function of $\mu$ and $h$ for a 1D disordered chain with $N = 400$ lattice sites, $t = 1$, $\alpha = 0.1\,t$, $\Delta = 0.05\,t$ and for different values of $\delta_\mu$. The topological region is drastically reduced only for values of $\delta_\mu$ greater than all relevant energy scales in the problem and comparable to the amplitude of the hopping, $t$. For very strong disorder, plotting the eigenvalues of the Hamiltonian (12) and the absolute square of the wave functions associated with MFs (numerically obtained as discussed in Section 4), we find that the spectrum is not gapped and the Majorana zero-energy modes are not localized at the edges of the chain, but completely overlapped (Figure 10).

The asymmetries, with respect to the $\mu = -2t$ axis, observable in Figure 9, are due to the fact that for all parameters, we always consider only one disorder configuration (and a chain of finite length, which is therefore not self-averaging). Indeed, plotting <Det(r)> vs. $N$ (Figure 11a) vs. different disorder configuration, $n_{dis}$, (Figure 11b), we observe that the transition from the trivial to the topological phase becomes sharper and symmetric with respect to the axis $\mu = -2t$, with increasing the system size or $n_{dis}$, respectively.

These results indicate that the topological phase is quite robust against disorder effects on the chemical potential. An additional test may be achieved considering the IPR [55,64]. As discussed in [64], the inverse participation ratio, which averages the fourth power of the wave function, allows us to distinguish the energy regions of extended and localized states of a quantum mechanical particle in a random potential. It is positive for localized states, and it vanishes for extended states in the thermodynamic limit. Figure 12) shows the IPR of the disordered chain (Equation (12)) for different values of $\delta_\mu$ and over a magnified energy interval in order to see the evolution close to zero energy, where MFs are expected to appear.

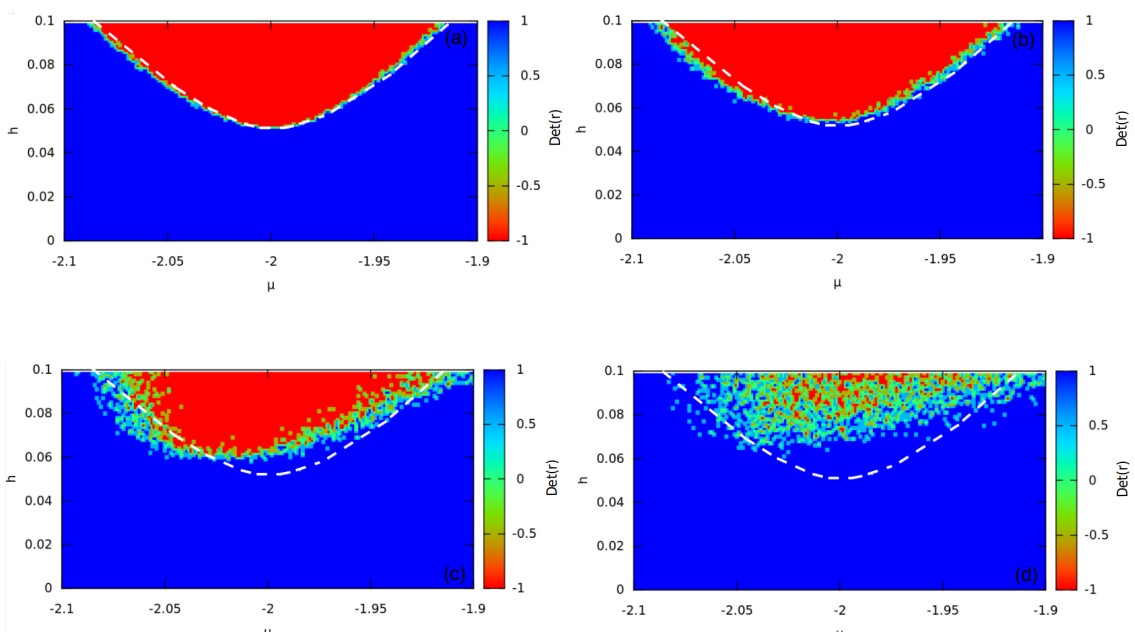

**Figure 9.** The determinant of the reflection matrix, $\mathrm{Det}(r)$, as a function of the chemical potential, $\mu$, of and the Zeeman field, $h$, in an inhomogeneous chain with $t = 1$, $\alpha = 0.1\,t$, $\Delta = 0.05\,t$, $N = 400$ and for different values of $\delta_\mu$. The topological region of the phase digram appears in red. Panel (**a**): $\delta_\mu/t = 0.1$; Panel (**b**): $\delta_\mu/t = 0.2$; Panel (**c**): $\delta_\mu/t = 0.4$; Panel (**d**): $\delta_\mu/t = 0.6$. The white dotted curve represents the critical field values, $h_c = \sqrt{\Delta^2 + (\mu + 2t)^2}$, for a clean chain.

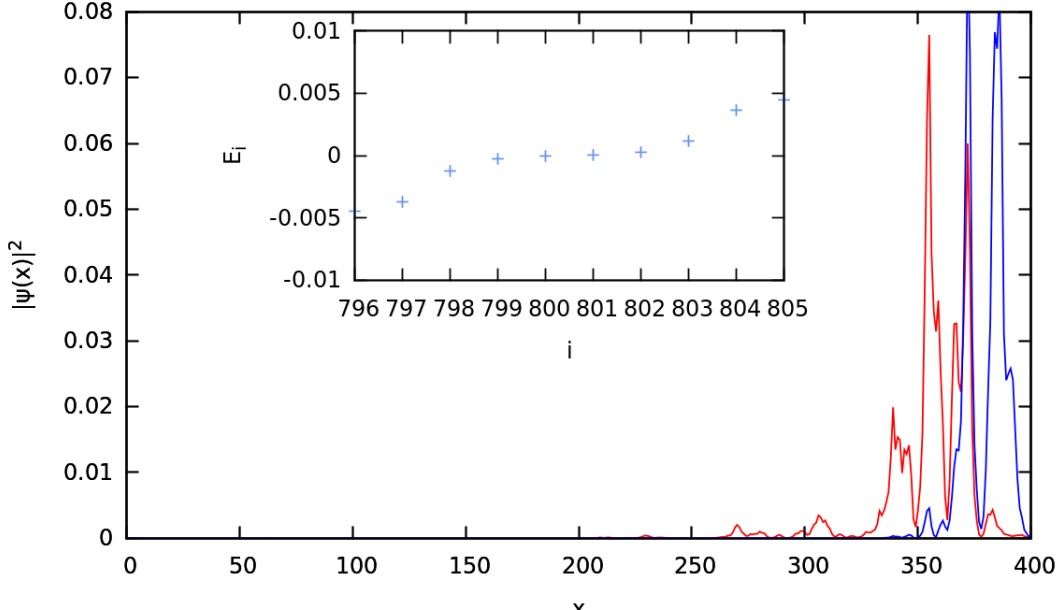

**Figure 10.** The MFs to vary the chain length with $t = 1$, $\alpha = 0.1\,t$, $\Delta = 0.05\,t$, $h = 0.09\,t$, $\delta_\mu = 0.66\,t$ for one disorder configuration. The figure shows the absolute square of the wave function $|\psi_+(x)|^2$ (red curve) and $|\psi_-(x)|^2$ (blue curve) (see Section 2). The inset shows the corresponding spectrum.

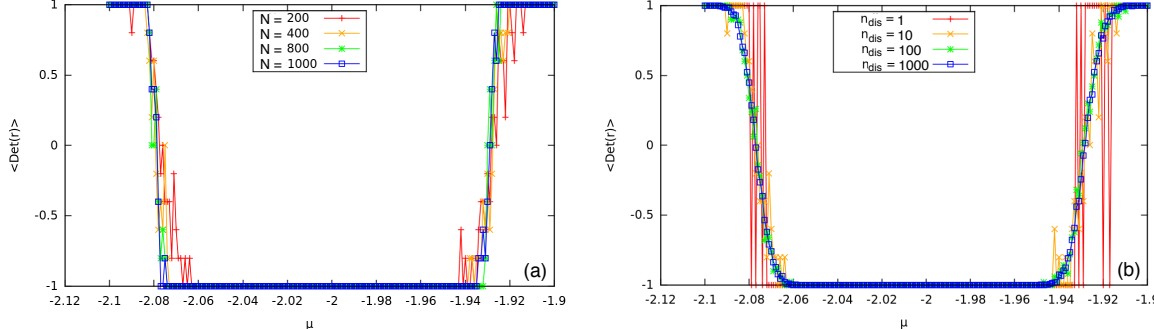

**Figure 11.** Panel (**a**): the mean value of Det($r$) calculated on $n_{dis} = 10$ disorder configurations for different system sizes and $t = 1$, $\alpha = 0.1\,t$, $\Delta = 0.05t$, $h = 0.09\,t$ and for $\delta_\mu = 0.2\,t$. Increasing $N$, the transition from the trivial to the topological phase becomes sharper, symmetrical with respect to the axis $\mu = -2t$, and fluctuations become weaker. Panel (**b**): the mean value of Det($r$) calculated for $N = 200$ and averaging over different numbers of disorder configurations $n_{dis}$, for the same parameters as in Panel (**a**). Increasing $n_{dis}$, fluctuations become weaker. The smearing of the transition from the trivial to the topological phase is a finite-size effect.

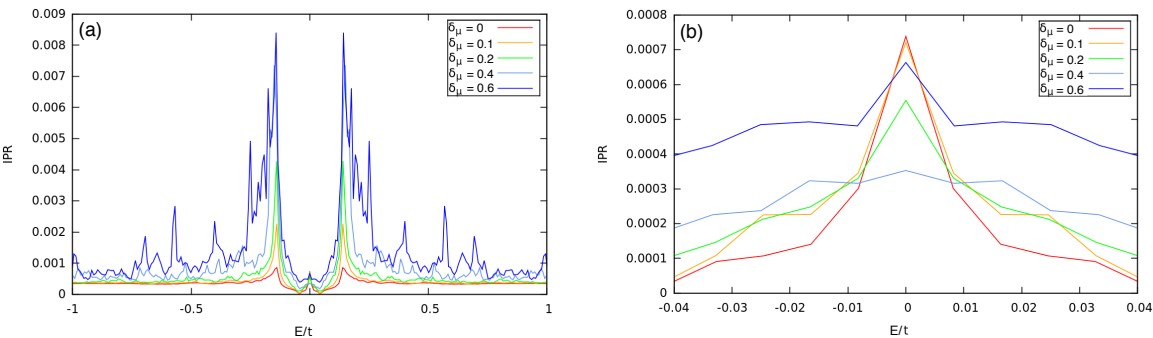

**Figure 12.** Panel (**a**): IPR for $n_{dis} = 1$, for an energy $E/t \in [-1 : 1]$, $N = 400$, $\alpha = 0.1\,t$, $\Delta = 0.05\,t$, $h = 0.09\,t$ and for different $\delta_\mu$ (in units of $t$). Panel (**b**): magnification for an energy $E/t \in [-0.04 : 0.04]$.

## 6. Discussion and Conclusions

The first part of this piece of work was devoted to a preliminary and rather pedagogical description of the topological properties of a model that is apt to capture the physics of a 1D structure that might be designed at the LAO/STO interface by means of cross top and back gating. Indeed, the joint presence of superconductivity [17–27], and of a strong RSOC [28–32], the large compressibility of the 2D electron gas [30,31,39–42] and the improvement of top gating [32] make these interfaces an ideal playground for detecting topological superconductivity [33,34].

The complementary study of the properties of the 1D model Hamiltonian in *k*-space (for periodic boundary conditions) and in real space (for open boundary conditions) allows us to fully characterize the conditions to reach the topological superconducting state and to discuss its properties. In particular, the real space analysis allows us to directly access the MFs that are found at the end-points of the chain in the topological phase and to characterize their properties, which is a prerequisite for the last part of this piece of work, where the effect of disorder on the topological superconducting state is analyzed, as recent studies show that MFs may suffer from strong disorder [50–57].

In the presence of defects or impurities, the crystal lattice is not perfectly periodic and the real space analysis is the main tool to investigate the properties of the system. The real space analysis must be complemented with the scattering matrix method, to provide information about the topological

properties of the disordered system. Aiming at describing 1D geometries with finite (meso- or nano-scopic) length, rather than relying on the self-averaging properties of a macroscopic disordered system, we mainly focused on the properties of the chain at a fixed disorder configuration, and we averaged over several disorder configurations only to check that the usual scenario is recovered in the case when self-averaging takes place, or to complement our study with the standard IPR criterion [64], as a further test to our theory.

Interestingly, we confirm that the topological properties are rather robust with respect to disorder. States well inside (or well outside) the topological region of the phase diagram are only affected by disorder when the strength of the disorder potential competes with the other relevant energy scales. On the other hand, there is a strip around the boundary of the topological region of the clean system where the fate of each individual state strongly depends on the disorder configuration, so that states may jump out of or in the topological phase, creating a very rough boundary between topological and non-topological states. The boundary becomes sharp only when an average is performed over several disordered configuration, or the size of the system is increased. Both effects reduce the fluctuations. This means that in a real disordered nanoscopic chain, in the region of the phase diagram near the boundary of the topological region of the clean system, MFs may form or disappear under tiny changes of, say, the chemical potential (through the top-gating potential) or the magnetic field.

**Author Contributions:** Conceptualization, M.V.M., N.S., M.G., S.C.; methodology, M.V.M., N.S., M.G., S.C.; software, M.V.M., N.S., with contributions from M.G., S.C.; formal analysis, M.V.M., N.S., M.G., S.C.; writing, original draft preparation, M.V.M., M.G., S.C., with contributions from N.S.

**Funding:** M.G. and S.C. acknowledge financial support from Sapienza Università di Roma, Project No. RM11715C642E8370.

**Acknowledgments:** We acknowledge valuable discussions with N.Bergeal and J. Lesueur.

**Conflicts of Interest:** The authors declare no conflict of interest.

## Abbreviations

The following abbreviations are used in this manuscript:

| | |
|---|---|
| MFs | Majorana fermions |
| 1D | one-dimensional |
| 2D | two-dimensional |
| LAO/STO | $LaAlO_3/SrTiO_3$ |
| RSOC | Rashba spin-orbit coupling |
| IPR | inverse participation ratio |

## Appendix A. Scattering Matrix Method

The scattering theory of electron conduction provides a complete description of transport at low frequencies, temperatures and voltages, provided the electron-electron interactions can be neglected.

The scattering matrix in Equation (14) can be obtained in the most convenient way within the transfer matrix scheme. Rather than relating the amplitudes of ingoing and outgoing modes, the transfer matrix relates the mode amplitudes to the right and to the left of the disordered chain,

$$\begin{pmatrix} \psi_{R,out} \\ \psi_{R,in} \end{pmatrix} = M \begin{pmatrix} \psi_{L,in} \\ \psi_{L,out} \end{pmatrix}.$$

In a given disorder configuration, the scattering matrix and the transfer matrix provide equivalent descriptions of the disordered region. Generally, one prefers to work with the transfer matrix because it satisfies simpler (i.e., multiplicative) composition rules.

In order to obtain the explicit expression of $M$, starting from the Hamiltonian (12), we must solve the zero-energy Schrödinger equation [50,57,59–61,63,65,66] yielding:

$$\begin{pmatrix} \hat{t}_x^\dagger \Phi_x \\ \Phi_{x+1} \end{pmatrix} = M_x \begin{pmatrix} \hat{t}_{x-1}^\dagger \Phi_{x-1} \\ \Phi_x \end{pmatrix},$$

where:

$$M_x = \begin{pmatrix} 0 & \hat{t}_x^\dagger \\ -\hat{t}_x^{-1} & \hat{t}_x^{-1} \hat{w}_x \end{pmatrix}.$$

Here, $\Phi_x$ is a four-component vector of wave amplitudes on site $x$. The waves at the two ends of the chain ($x = 1$ and $N$) are related by the transfer matrix [50,57] $M = M_N M_{N-1}, ..., M_2 M_1$.

We transform to a new basis with right-moving and left-moving electrons separated in the upper and lower four components, by means of the unitary transformation:

$$\tilde{M} = U^\dagger M U, \quad U = \frac{1}{\sqrt{2}} \begin{pmatrix} 1 & 1 \\ \iota & -\iota \end{pmatrix}.$$

In this basis, the transmission and reflection matrices follow from the relations:

$$\begin{pmatrix} t \\ 0 \end{pmatrix} = \tilde{M} \begin{pmatrix} 1 \\ r \end{pmatrix}, \quad \begin{pmatrix} r' \\ 1 \end{pmatrix} = \tilde{M} \begin{pmatrix} 0 \\ t' \end{pmatrix}.$$

Unitarity, together with particle-hole symmetry, ensures that the determinants of $r$ and $r'$ are identical real numbers.

This recursive construction is numerically unstable, because products of transfer matrices contain exponentially-growing eigenvalues, which overwhelm the small eigenvalues relevant for transport properties. In order to calculate $\tilde{M}$, then, we use the numerical scheme introduced by Snyman et al. [67] and described in detail in the work of Nori et al. [50]. In this method, using the current conservation condition, the product of transfer matrices is converted into a composition of unitary matrices, involving only eigenvalues of unit modulus.

Exploiting this formalism, we recover the result of Section 3 for a clean chain. Figure A1 reports Det($r$) and it clearly shows that the topological phase (red region) occurs for values of $h$ fulfilling the topological condition Equation (8) (represented by the dashed withe line). We also find that in the clean case, no effect is played by the RSOC, in agreement with the finding that the topological criterion does not depend on this parameter.

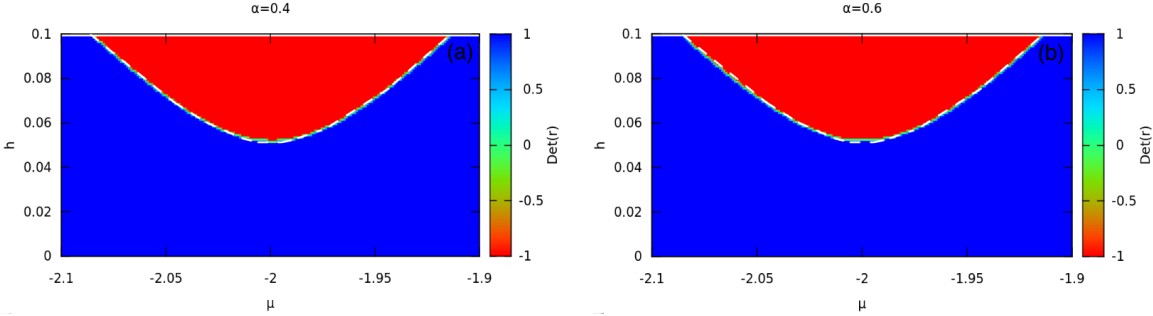

**Figure A1.** The determinant of the reflection matrix, Det($r$), as a function of the chemical potential, $\mu$, and the Zeeman field, $h$, in a clean chain with $t = 1$, $\Delta = 0.05\, t$, $N = 400$. Panel (**a**): $\alpha = 0.4\, t$; Panel (**b**): $\alpha = 0.6\, t$.

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
