# Peer review of "Majorana Fermions in One-Dimensional Structures at LaAlO3/SrTiO3 Oxide Interfaces"

_condensedmatter, doi:10.3390/condmat3040037_

Round 1

Reviewer 1 Report

In the manuscript, M. V. Mazziiotti et al. reported a theoretical calculation to predict the topological superconducting phase and as well as Majorana end states in a one-dimensional structure at the LaAlO3/SrTiO3 interface. By assuming a Rashba type spin orbit coupling and s-wave superconducting in the heterostructure, the authors found that the system could be described by an effective Kitaev model. Under a certain range of the applied in plane magnetic field, the system can be changed into a topological non-trivial phase with Majorana zero modes on its boundaries. I found the work is comprehensive and technically robust. I recommend the manuscript for publication after the authors carefully check the manuscript to correct the typos e.g. “shll” in line 166.

Author Response

We thank the referee for her/his appreciation of our work, and for  her/his careful reading of our manuscript. We spotted few more typos, that were all corrected in the revised version.

We hope that our paper is now suitable for publication in Condensed Matter.

Reviewer 2 Report

I recommend publication of the manuscript if the authors provide adequate responses to the following issues:

At what point is this model different from the models used for e.g. semiconductor nanowire structures? (apart from minor tweaks in the paramaters)

Does the model apply to LAO-STO? It is single band; superconductivity in LAO-STO seems to be only present in the dxz/dyz bands (e.g. DOI: 10.1038/s41467-018-02907-8). Also the model supposes a parabolic band with spin orbit coupling, whereas LAO-STO's Rashba effect is linked to both multiband structure and deviations from parabolic behavior. (e.g., ref 37)

The previous work on majorana fermions in LAO-STO needs to be discussed. https://doi.org/10.1209/0295-5075/108/60001

Author Response

We thank the referee for her/his appreciation of our work, remarks and constructive criticism. We followed her/his suggestions and modified our manuscript accordingly. Hereafter, we discuss point by point the report.

1. Our model is indeed a model for a gate-induced “nanowire” geometry, which takes into account the peculiarities of the electron gas at the LAO/STO interface, namely, intrinsic s-wave superconductivity (which therefore does not require any proximity-induced superconductivity) and Rashba spin-orbit coupling. We agree with the referee that this point can be made clearer, and we added a sentence on this aspect when the model is introduced in Sec. 2.

2. The referee is perfectly right in a two-dimensional geometry. In a one-dimensional confined geometry the dyz band is pushed up in energy by confinement in the y direction. The same occurs to the dxy band. Whether this becomes degenerate with the dxz depends on the degree of confinement along the y and z directions. We understand that our discussion on this aspect in the first two paragraphs of Sec. 2 was not sufficiently clear and we tried to improve it.

3. We really apologize for missing the quotation to the paper by Mohanta and Taraphder (MT) which, by the way, has inspired our work. We do not know how this relevant reference somehow disappeared from the final version of our manuscript and we restore it in the revised version. As far as the comparison between the two approaches is concerned, we can say that: 

a. The MT model is a single-band model for a two-dimensional electron gas, whereas we deal with a confined one-dimensional single-band model. 

b. MT assume the presence of ferromagnetic moments as a property of the LAO/STO interface. This point of view, which was fully legitimate in 2014, does not seem to be supported by more recents results, and therefore we assume that the magnetic field along x is externally applied.

c. MT need an extra magnetic field along z to push one of the chiral states to higher energy, thereby reaching an effective spinless state. This is not required in our one-dimensional geometry. In both cases, the superconducting states acquires an effective p-wave character thanks to the Rashba spin-orbit interaction.

d. Despite some similarity in the expressions for the effective s-wave and p-wave superconducting order parameters, the different roles of the magnetic fields is such that for us hx favors the topological state, contrary to what found by MT. Of course, both results are meaningful within the different assumptions.

e. Our analysis of disorder effects, which is the main result of our work, is clearly tailored for the one-dimensional geometry we are considering. 

We have introduced in Sec. 2 a brief comparison with the approach of MT, as well as proper reference to their work.

We hope that, having taken into account all the suggestions of the referee, our paper is now suitable for publication in Condensed Matter.

Round 2

Reviewer 2 Report

I recommend publication